

# Simulating Floquet topological phases in static systems

**Selma Franca[1]⋆, Fabian Hassler[2] and Ion Cosma Fulga[1]**

**1** IFW Dresden and Würzburg-Dresden Cluster of Excellence ct.qmat,
Helmholtzstr. 20, 01069 Dresden, Germany
**2** JARA-Institute for Quantum Information, RWTH Aachen University,
52056 Aachen, Germany

⋆ s.franca@ifw-dresden.de

## Abstract

We show that scattering from the boundary of static, higher-order topological insulators (HOTIs) can be used to simulate the behavior of (time-periodic) Floquet topological insulators. We consider $D$-dimensional HOTIs with gapless corner states which are weakly probed by external waves in a scattering setup. We find that the unitary reflection matrix describing back-scattering from the boundary of the HOTI is topologically equivalent to a $(D-1)$-dimensional nontrivial Floquet operator. To characterize the topology of the reflection matrix, we introduce the concept of 'nested' scattering matrices. Our results provide a route to engineer topological Floquet systems in the lab without the need for external driving. As benefit, the topological system does not suffer from decoherence and heating.

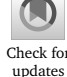

# 1  Introduction

Topology provides a common tool set for analyzing the properties of both static and dynamical systems. Bulk-boundary correspondence predicts the appearance of gapless modes both in the spectra of time-independent Hermitian Hamiltonians [1, 2], as well as in those of the unitary time-evolution (or Floquet) operators describing periodically-driven systems [3, 4]. Both Hermitian and unitary phases have been classified using dimensional reduction, leading to the so called 'periodic tables' of topological phases [5–15].

In spite of the common methods used in their analysis, there is a strong dichotomy between the study of topology in Hermitian and unitary systems, both on a theoretical and especially on an experimental level. Topological phases described by Hamiltonians are usually realized in the ground states of isolated, time-independent systems [1,2], and a large body of research has focused on materials that exhibit topologically nontrivial ground states [16–22]. In contrast, unitary topology conventionally refers to Floquet operators that physically involve pumping energy into an open system by means of an external driving field. In the many-body setting, it is known that the system will absorb this energy, reaching a featureless steady state unless it is many-body localized [23, 24]. Even on the single-particle level, heating due to noise-induced decoherence of Floquet states is unavoidable, since any realistic driving field will not be perfectly periodic in time [25–27].

In this work, we resolve the dichotomy between Hermitian and unitary systems by showing that a static, $D$-dimensional topological phase can be used to simulate a time-periodic, $(D-1)$-dimensional topological phase, without any external driving. We consider a sub-class of higher-order topological insulators (HOTIs) [28–34]: time-independent systems with a gapped bulk, gapped boundaries, and topologically protected gapless modes localized at their corners. We envision a scattering experiment on the $D$-dimensional HOTI by probing the system at the boundary. Due to the gap, all incoming modes are back-reflected, a process described by a reflection matrix $r$ (see Fig. 1). Our main insight is that this reflection matrix: (1) is unitary, (2) $(D-1)$-dimensional, (3) has a gapped spectrum, and (4) shows topologically protected mid-gap modes at its boundaries. As such, the unitary reflection matrix can be thought of as the Floquet operator of a lower-dimensional driven system.

On a practical level, our results provide a way of experimentally realizing unitary topological phases in a static experiment. The required ingredients, a HOTI probed by means of a scattering measurement, are presently available in the lab. HOTIs have been achieved in a variety of metamaterials [35–50]. The reflection matrix can be determined by standard interferometric techniques [51, 52], or by simply visualizing the standing wave pattern formed between incoming and outgoing modes [49, 53]. Note that alternative proposals for simulating Floquet phases, such as photonic crystals [54–57] or quantum walks [58, 59], suffer from

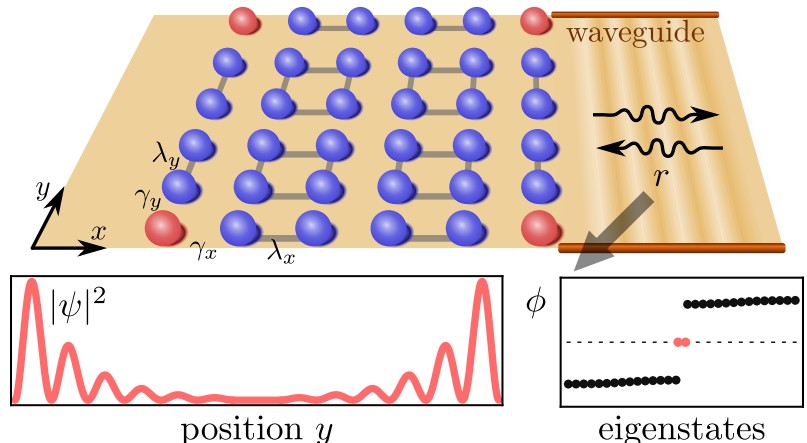

Figure 1: Scattering experiment in which a waveguide is attached to the right boundary of a HOTI [Eq. (1)] with corner states (red). Due to the gapped bulk and edges, the incoming waves are reflected. This process is described by the unitary reflection matrix $r$, that is topologically equivalent to the Floquet operator of a nontrivial 1D chain. The spectrum of its eigenphases $\phi$, denoting the phase difference between incoming and reflected waves, exhibits topologically protected mid-gap states. The corresponding transversal modes $\psi$ are localized at the boundaries of the waveguide and scatter off the corners of the 2D system.

decoherence due to noise in the periodic modulation of the system. In contrast, our setup has the advantage of avoiding decoherence. This is because there is no driving field to begin with: the HOTI is fully static and remains in its ground state, being only weakly probed.

We first present our main idea in a generic setting (Sec. 2) before proceeding with a concrete example of a two-dimensional (2D) particle-hole symmetric HOTI. For the latter system, we show that the reflection matrix is topologically equivalent to a 1D Floquet Kitaev chain [60–62], realizing the same topological phases (Sec. 3). By interpreting the reflection matrix as a 1D Floquet operator and computing its scattering matrix (e.g., the scattering matrix of the reflection matrix) [63, 64], we determine the topological invariants (Sec. 4). This defines a generic, recursive procedure similar to that of 'nested Wilson loops' [28, 29]. It enables to access the unitary topology encoded in a scattering matrix by computing its higher-order, nested scattering matrices. We show that the resulting topological phases are robust against disorder as long as their protecting symmetries are preserved (Sec. 5). We argue that the topological phases can be identified using experimental tools which have already been demonstrated (Sec. 6). We conclude and discuss future directions of research in Sec. 7. The appendix is dedicated to details on how we compute the scattering matrix (App. A), its symmetries (App. B), the dimensional reduction map (App. C), as well as to a discussion on chiral symmetry (App. D), topological phase transitions of the reflection matrix (App. E), and the connection to Floquet invariants (App. F).

## 2 Main idea

Consider the 2D HOTI shown in Fig. 1, which has gapped bulk and edges, shown in blue, but hosts topologically protected gapless corner modes, shown in red. The right edge of this static system, having a linear dimension of $L$ sites, is weakly coupled to a waveguide with a set of $L$ modes that are used to probe the system. Since the bulk and the edges of the HOTI are fully gaped, no transmission occurs through the system, and all incoming waves are back-reflected.

Therefore, the reflection matrix $r$, whose elements $(r)_{nm}$ describe the probability amplitude to scatter from (incoming) mode $m$ to (outgoing) mode $n$, is forced to be unitary. The eigenvalues $e^{i\phi}$ of this matrix determine the eigenphases $\phi$ acquired by modes upon reflecting from the system edge.

Consider an incoming plane-wave, extended along the translationally invariant direction of the waveguide, but localized in its transversal direction $y$ (see Fig. 1). The phase it acquires upon reflection will depend on the transversal direction. If the reflection occurs from a portion close to the middle of the gapped edge, we expect the outgoing state to pick up an eigenphase $\phi$, which may depend on the properties of the edge close to the position $y$ and its coupling to the waveguide. In fact, for the choice of system and waveguide considered in the following, we have an open boundary condition with $\phi \to 0$ for vanishingly weak coupling strength. In contrast, if the incoming plane-wave is localized at the boundary of the waveguide, such that it impinges on the corner of the HOTI, then resonant scattering from the gapless corner mode will force the outgoing state to pick up a $\pi$-phase relative to the incoming one [65–68]. As a result, the eigenstates of $r$ at the center of the waveguide have eigenphases $\phi$ close to zero, and there is one eigenvalue with $\phi = \pi$ located at each of the two boundaries of the waveguide. The reflection matrix of this 2D system thus corresponds to a 1D topological Floquet chain (along the transversal direction) with mid-gap topological modes (at $\phi = \pi$) localized at the ends of the chain. Note that the $\pi$-modes are specific to the classification of unitary Floquet systems and do not have an analog for Hamiltonian systems. Thus, the process of weakly probing a static HOTI is determined by the reflection matrix $r$, which is equivalent to a Floquet operator. If the static system is in the HOTI phase, the resulting Floquet operator is also topological; this is the sense in which the static system simulates a topological Floquet system.

By analogy, we expect the connection between the topology of the Hamiltonian and that of the reflection matrix to remain valid in arbitrary dimension: a $D$-dimensional HOTI in which the bulk and hypersurfaces are gapped, but which hosts gapless 0D corner modes at its $2^D$ corners, will have a reflection matrix that simulates a $(D-1)$-dimensional topological Floquet system. As in the 2D example of Fig. 1, this is because waves which are back-scattered from the corners will produce $\pi$-modes in the reflection matrix, whereas waves reflected from the middle of the surface of the system will produce modes with $\phi \to 0$ in the weak-coupling limit. We make these arguments precise in Appendix C.

## 3   Hamiltonian and scattering matrix

To make the above discussion concrete, we consider a HOTI model describing non-interacting, spinless fermions hopping on a dimerized square lattice [28,29]. There are four sites per unit cell, and each plaquette is threaded by a magnetic $\pi$-flux. The Hamiltonian reads

$$h(\boldsymbol{k}) = (\gamma_x + \lambda_x \cos k_x)\tau_x\sigma_0 - \lambda_x \sin k_x \tau_y\sigma_z - (\gamma_y + \lambda_y \cos k_y)\tau_y\sigma_y - \lambda_y \sin k_y \tau_y\sigma_x, \quad (1)$$

where $\boldsymbol{k} = (k_x, k_y)$ are the momenta in the two directions. The Pauli matrices $\tau$ act on the sublattice degree of freedom, whereas the Pauli matrices $\sigma$ act on the two sites within a sublattice. Similar to the Su-Schrieffer-Heeger (SSH) model [69], the hoppings in the horizontal and vertical ($x$ and $y$) directions are dimerized, taking values $\gamma_{x,y}$ within a unit cell and $\lambda_{x,y}$ between unit cells, all chosen real and positive in the following. The model obeys particle-hole symmetry, $\mathcal{P} = \tau_z \mathcal{K}$, with $\mathcal{K}$ complex conjugation, such that all states are symmetric in energy around $E = 0$.[1] For $\gamma_{x,y} < \lambda_{x,y}$ the system is in a topological phase: the bulk/edges

---

[1]Note that the system has an additional chiral symmetry, $\mathcal{C} = \tau_z$, which has no effect as we do not consider more than a single 0- or $\pi$-mode. For more information see Appendix D.

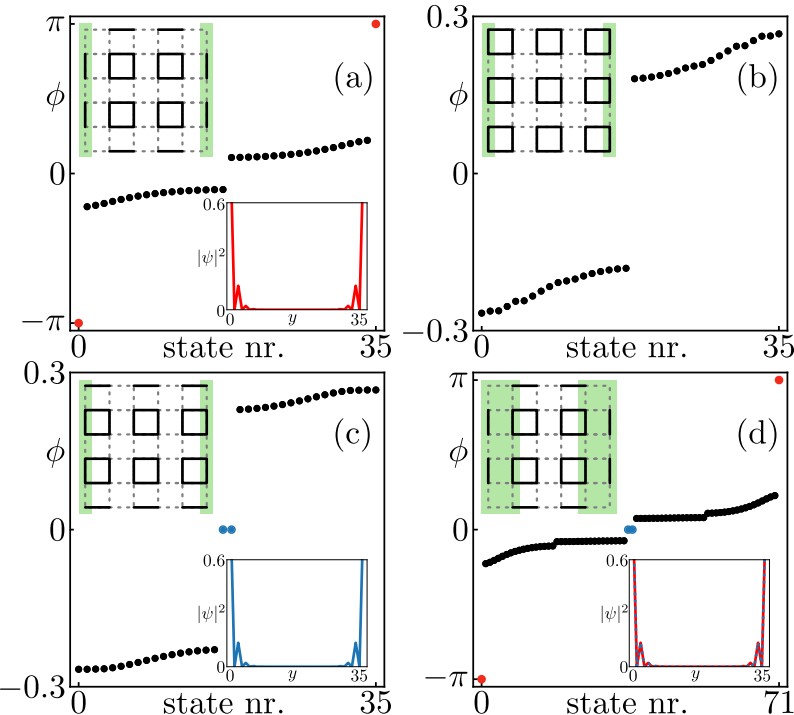

Figure 2: Eigenphases of the reflection matrix, showing $\pi$-modes (0-modes) and their associated wavefunctions in red (blue). The HOTI parameters are $\lambda_{x,y} = 1$ in all panels. We have chosen $\gamma_{x,y} = 0.4$ in panels (a) and (d), $\gamma_{x,y} = 1.2$ in panel (b), whereas $\gamma_x = 1.2$ and $\gamma_y = 0.4$ in panel (c). Insets sketch the corresponding dimerization pattern of the HOTI as well as the sites connected to the lead (green). The system-lead coupling strength is $t_{sl} = 0.5$ in all cases. The gap around $\phi = 0$ as well as the bandwidth of the bulk states (black) is proportional to $t_{sl}$ in the weak coupling limit.

are gapped and corner sites are weakly coupled to the rest of the system (see Fig. 1) such that they each host a localized, zero-energy mode.

We place a finite-sized HOTI ($L \times L$ sites with $L = 36$) in a two-terminal geometry with translationally invariant electronic waveguides (leads) oriented in the $x$ direction attached to its left- and right-most sites. Incoming and outgoing modes at $E = 0$ are related by the scattering matrix,

$$S = \begin{pmatrix} r & t' \\ t & r' \end{pmatrix}, \tag{2}$$

where the blocks $r^{(\prime)}$ and $t^{(\prime)}$ contain the probability amplitudes for states to be back-reflected or transmitted across the system, respectively. More details on calculating $S$ can be found in Appendix A. To obtain direct information on the real-space position of a scattering state in the $y$ direction, we choose a simple model for the leads composed of decoupled chains (zero on-site term, unit hopping), each chain connecting to a single site of the edge of the system. As a result, the reflection and transmission blocks have a size $L \times L$ and are effectively 1D operators parameterized by the transversal coordinate $y$.

In the nontrivial phase, transmission through the system is exponentially suppressed since the bulk and edges are gapped. Hence, the reflection matrix $r$ of the left lead (or equivalently $r'$ for the right lead) is unitary up to exponential precision. By numerically[2] diagonalizing the

---

[2]We use the kwant code for tight-binding and scattering matrix calculations [70]. The code used to generate our numerical results is included in the supplemental material.

1D unitary operator $r$, we find that its eigenphase spectrum is analogous to the eigenphase (or quasi-energy) spectrum of 1D Floquet topological chains, as shown in Fig. 2a. We observe two $\pm\phi$-symmetric phase-bands, shown in black, separated by phase-gaps centered around $\phi = 0$ and $\phi = \pi$. The bands correspond to states which are back-reflected from the mid section, or 'bulk' of the lead, meaning states which contact the central part of the HOTI edge. At $\phi = \pm\pi$, however, there are two degenerate states, shown in red, which are separated by a gap from other eigenphases. These two modes correspond to scattering states at the boundaries of the lead and contact the corner states of the HOTI.

The reflection matrix $r$ is topologically equivalent to a 1D nontrivial Floquet system and thus provides an example of dimensional reduction from a 2D Hermitian operator to a 1D unitary operator. The latter is in a symmetry class allowing for nontrivial topology, since it inherits particle-hole symmetry from the original 2D HOTI. The $\mathcal{P}$ symmetry of the Hamiltonian and of the leads implies a constraint on the reflection matrix, $r = \tau_z r^* \tau_z$ [67,71], which is identical to the condition imposed by particle-hole symmetry on Floquet systems (see Refs. [3,63] and Appendix B). Due to the constraint relating $r$ to its complex conjugate, the eigenvalues of $r$ must be either real, corresponding to phases $\phi = 0, \pi$, or must come in complex conjugate pairs. This explains both the $\pm\phi$-symmetric spectrum of Fig. 2 and the pinning of edge modes to the middle of the gap. Furthermore, since resonant scattering from a zero-energy state produces a $\pi$ phase shift of the reflected wave [67], the corner states of a nontrivial HOTI induce topological $\pi$-modes in the reflection operator. These modes are associated with a quantized topological response indicative of the nontrivial nature of $r$: the phase difference between incoming and outgoing waveguide modes is quantized to $\pi$ when backscattering occurs at the waveguide boundaries. Therefore, the dimensional reduction scheme obeys the requirement of mapping a nontrivial system onto a nontrivial one (see Appendix C for details on this dimensional reduction map).

Particle-hole symmetric Floquet systems in 1D possess a $\mathbb{Z}_2 \times \mathbb{Z}_2$ classification [72], with four possible phases: (1) trivial, (2) edge modes at $\phi = \pi$, (3) edge modes at $\phi = 0$, and (4) a so-called 'anomalous' phase hosting edge modes at both 0 and $\pi$. Depending on the dimerization pattern of the 2D system and on how the leads are attached, we show that all four phases can be reproduced by the reflection matrix. As an immediate check, a trivial system, $\gamma_{x,y} > \lambda_{x,y}$, yields a trivial $r$ (Fig. 2b). The phase with $\phi = 0$ modes (Fig. 2c) can be obtained by setting $\gamma_x > \lambda_x$ and $\gamma_y < \lambda_y$ (nontrivial dimerization on the edge contacting the leads, but trivial dimerization along the other edges). In this situation, the system is in a topological phase (as signaled by its nontrivial nested Wilson loop invariant [28]) even though it does not exhibit zero-energy corner states. Remarkably, the reflection matrix detects the topology of the phase even in this case, by the presence of the $\phi = 0$ modes.

So far, the waveguides were only coupled to the outermost sites of the HOTI. Thus, only half of the sites of the last unit cell are coupled to the lead, since the unit cell contains four sites and only two of them form the boundary of the system (see Fig. 1). However, depending on the physical system realizing the HOTI phase, the matrix structure of the Hamiltonian Eq. (1) might be due to internal degrees of freedom. For instance, the Pauli matrices $\tau$ and $\sigma$ could represent, e.g., electron-hole and spin degrees of freedom in a superconductor. In this case, a lead would couple to all four states in a unit cell, producing a reflection matrix which is $2L \times 2L$. Interestingly, attaching the lead in this way maps the nontrivial HOTI ($\gamma_{x,y} < \lambda_{x,y}$) to an anomalous $r$, with mid-gap end modes at both $\phi = 0$ and $\phi = \pi$ (Fig. 2d). Even though the system Hamiltonian is in the same nontrivial phase as was used to produce Fig. 2a, $r$ is now in a different topological phase. The difference is due to the fact that, both for static and for Floquet chiral systems [28, 29, 69, 73–77], the topology is not just a property of the HOTI Hamiltonian, but crucially depends on the way in which the system is terminated. The important point however, as discussed before, is that zero-energy corner states in the 2D HOTI

lead to $\pi$ modes in the reflection matrix, such that a nontrivial static system is mapped onto a nontrivial unitary Floquet operator.

Finally, we make some remarks on the topology of the reflection matrix in the context of previous HOTI classification studies. According to Refs. [33,34], HOTIs fall within two large classes. In *intrinsic* HOTIs, the zero-energy corner states are associated with a bulk topological invariant which is protected by lattice symmetries. This is the case for the Hamiltonian in Eq. (1), where the bulk invariant relies on fourfold rotation symmetry and has been computed in Ref. [29]. In *extrinsic* HOTIs, however, the bulk is trivial and corner states at $E = 0$ are a consequence of the fact that the system's boundary is in a strong topological phase. By weakly breaking the lattice symmetries while preserving the particle-hole (or the chiral) symmetry, the Hamiltonian Eq. (1) transitions from an intrinsic to an extrinsic HOTI phase. The bulk becomes trivial due to the breaking of the symmetries required to define the invariant, but mid-gap corner states remain protected due to the strong topology of the edge. This can in fact be seen in Fig. 1: the edges of the system are nontrivial SSH chains. Therefore, the corner modes of the 2D system and the topological modes of its reflection matrix do not require lattice symmetries for their protection. We explore this fact later in Section 5, as well as in Appendix C, in which we discuss the range of validity of our dimensional reduction scheme.

## 4  Nested scattering matrices and topological invariants

We want to prove that the reflection matrix $r$ of the HOTI is indeed in a topological phase when viewed as a unitary Floquet operator by computing its topological invariants. The spectra shown in Fig. 2 give a first indication that this is the case, due to the presence of mid-gap modes. The most conventional way of calculating the invariants of a periodically-driven system [9, 78] relies on access to the instantaneous eigenstates of the system at every moment of time throughout a period of the drive. In our system, we do not have access to these instantaneous eigenstates, but only to the reflection matrix $r$, which is analogous to the Floquet operator — the time-evolution operator over a full driving cycle. One possibility is to mimic a time-evolution process by finding a continuous way of unitarily deforming the reflection matrix to the identity operator, which we explore in Appendix F. In the following, we rely instead on a recently developed way of determining the topological invariants of a 1D Floquet system [63, 64], which only uses knowledge of $r$ (the analog of the Floquet operator). To this end, we calculate the scattering matrix $\tilde{S}$ of a 1D Floquet system described by the 'Floquet operator' $r$. We call $\tilde{S}$ a nested scattering matrix, as it is the scattering matrix of the reflection matrix $r$ (which is a sub-block of the scattering matrix $S$).

Following Ref. [63], the procedure to obtain the topological invariants is as follows: we define a fictitious scattering problem starting from the finite-sized, 1D unitary operator $r$. We attach one fictitious absorbing terminal to the first ($y = 0, 1$) and one to last ($y = L-2, L-1$) unit cell of the 1D system. The projection operator onto the absorbing terminals,

$$P = \begin{cases} 1, & \text{if } y \in \{0, 1, L-2, L-1\}, \\ 0, & \text{otherwise.} \end{cases} \tag{3}$$

is of size $4 \times L$ or $8 \times 2L$, depending on whether the waveguides probing the HOTI are attached only to the last sites (as in Fig. 2a,b,c), or to the full unit cell (as in Fig. 2d). Transmission $\tilde{t}^{(\prime)}$ and reflection $\tilde{r}^{(\prime)}$ from the two absorbing terminals at the ends of the 1D system can be computed from the nested scattering matrix[3]

$$\tilde{S}(\phi) = P[1 - e^{i\phi} r(1 - P^T P)]^{-1} e^{i\phi} r P^T, \tag{4}$$

---

[3]The equation is identical to that used in Refs. [63] to study driven systems, with the exception that the Floquet operator $\mathcal{F}$ has been replaced by $r$, and its quasi-energies $\varepsilon$ have been replaced by the phases $\phi$.

having the same block structure as Eq. (2). The interpretation of Eq. (4) in Floquet language is as follows: it corresponds to an infinite sum over different scattering processes, obtained by expanding the inverse in a geometric series. Each successive term describes time-evolution over an additional period, obtained by applying the 'Floquet operator' $r$. Each state is projected out if it reaches the absorbing terminals (given by $P$) and continues evolving for another driving cycle if it does not overlap with the terminals (given by $1-P^T P$). The nested scattering matrix $\tilde{S}$ inherits particle-hole symmetry from the reflection block of the scattering matrix $S$. Indeed, $r = \tau_z r^* \tau_z$ together with Eq. (4) imply $\tilde{r}(\phi) = \tau_z \tilde{r}^*(-\phi)\tau_z$ [63]. As such, the determinants of $\tilde{r}(\phi = 0)$ and $\tilde{r}(\phi = \pi)$ are real and the topological invariants are given by

$$\nu^0 = \operatorname{sign} \det[\tilde{r}(0)], \qquad\qquad \nu^\pi = \operatorname{sign} \det[\tilde{r}(\pi)], \qquad\qquad (5)$$

see also [60, 67, 68, 79, 80]. We have checked that the two $\mathbb{Z}_2$ indices correctly capture the presence of topological modes in the spectrum of $r$. Whenever there are mid-gap modes at $\phi = 0$ or $\phi = \pi$, we find that $\nu^0$ or $\nu^\pi$ take the nontrivial value $-1$. Otherwise, they take the trivial value $+1$. In Appendix E, we show how changing the HOTI parameters leads to topological phase transitions of the reflection matrix, signaled by changes in its topological invariants. These phase transitions can occur both by preserving the unitarity of $r$ (the 2D HOTI remains insulating), or by rendering it singular (the 2D HOTI conducts).

## 5 Disorder effects

Systems with gapless corner modes have successfully been realized in many different platforms [35–50], most of them being the classical analogs of condensed matter systems. Even though such realizations offer a better control over disorder, its complete elimination remains a challenging task. For this reason, we consider two types of imperfections in the 2D system. One kind of disorder breaks the particle-hole symmetry, while the other one preserves it. In the following, we will study how these disorder types affect topological phases of the reflection matrix.

Disorder which breaks $\mathcal{P}$ is simulated by means of random onsite energies, drawn independently for every site from the uniform distribution on $[-d_{\text{ons}}, d_{\text{ons}}]$. Here, $d_{\text{ons}}$ denotes the strength of disorder. Disorder which preserves $\mathcal{P}$ is modeled with $\gamma_{x(y)} \to \gamma_{x(y)} + \delta_{\text{hop}}$ and $\lambda_{x(y)} \to \lambda_{x(y)} + \delta_{\text{hop}}$, where $\delta_{\text{hop}}$ drawn independently for each hopping from the uniform distribution on $[-d_{\text{hop}}, d_{\text{hop}}]$, where $d_{\text{hop}}$ represents disorder strength. For both situations, we consider 250 disorder realizations and study averaged density of states (DOS) of the eigenphases of the reflection matrix. The results for phases with 0-modes and $\pi$-modes are given in Fig. 3.

The effect of the onsite disorder that breaks $\mathcal{P}$ on 0-modes and $\pi$-modes is shown in Fig. 3a and Fig. 3b, respectively. We expect this type of disorder to shift the eigenphases of end modes from the particle-hole (and chiral) invariant points of the $\phi$-spectrum, thus diminishing the DOS peak at $\phi = 0, \pi$. This effect is noticeable in both panels, and even for very small disorder strengths. Further increase of the disorder strength pushes eigenphases of these end states (at $\phi = 0, \pi$ in the clean limit) towards the bulk values, after which the DOS midgap peak disappears. Finally, even though $\pi$-modes are related to gapless Hamiltonian states, while 0-modes are not, both kinds are similarly affected by disorder. This is because the existence of both kinds of states originates from symmetry requirements on $r$.

On the other side, we expect the nontrivial phase of the reflection matrix to be more stable to spatial disorder that preserves $\mathcal{P}$. This is because for this type of disorder the 2D system enters an extrinsic HOTI phase, as discussed above. The disorder cannot move corner states from zero-energy but it reduces the bulk mobility gap. Once a mobility gap closing occurs, the

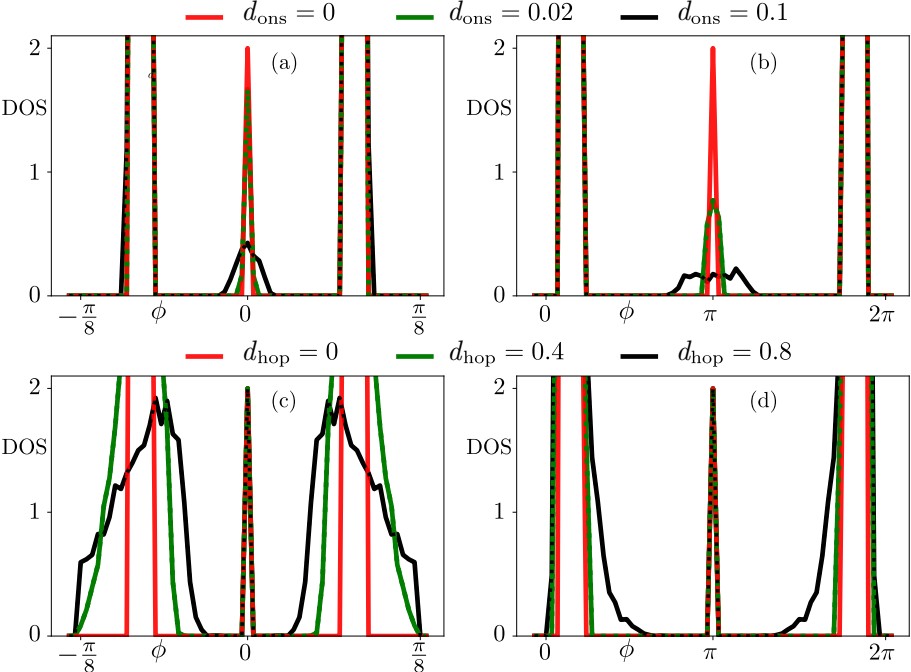

Figure 3: The evolution of the DOS (of the reflection matrix) with disorder. For every disorder realization, we calculate the DOS by dividing the full eigenphase range into 60 equally sized intervals. Panels (a) and (b) concern onsite disorder, while panels (c) and (d) reveal effects of randomness in hopping strengths. In panels (a) and (c), we consider a reflection matrix with 0-modes ($\gamma_x = 1.2, \gamma_y = 0.4, \lambda_x = \lambda_y = 1$), while panels (b) and (d) show how disorder influences a reflection matrix with only $\pi$-modes ($\gamma_x = \gamma_y = 0.4, \lambda_x = \lambda_y = 1$). In all cases, the system size is $60 \times 60$ sites.

reflection matrix loses unitarity, and cannot be used to simulate Floquet phases of Hermitian systems. For this reason, we restrict ourselves to disorder strengths which leave $r$ unitary for every disorder realization with $1 - |\det r| \leq 10^{-6}$. As randomness in hoppings preserves $\mathcal{P}$, 0-modes and $\pi$-modes remain pinned to $\phi = 0, \pi$ as observed in Fig. 3c and in Fig. 3d, respectively.

## 6 Experimental feasibility

Our insight that a static system which is weakly probed simulates a Floquet system opens an alternate route to the experimental realization of Floquet topological phases. The method relies on two parts: (1) constructing a HOTI and (2) measuring the eigenphases of its reflection matrix. The first part has already been achieved in photonic [35–37], phononic [38], microwave [39], acoustic [41–47, 50], topoelectric [40], and condensed-matter [49] metamaterials, producing HOTI Hamiltonians similar to Eq. (1). Many of these platforms also allow to directly measure the phase of reflected waves, either by means of interferometry, as done in microwave [51], photonic [52], acoustic [81], and topoelectric [82] systems, or by directly visualizing the standing wave pattern formed at the sample boundary, as done, e.g., with electronic [49] or water waves [53]. Indeed, the authors of [49] have recently reported experiments realizing a HOTI by placing impurities on a surface in a Kagome lattice and measuring the system using scanning tunneling microscopy. While the focus of their paper is on the system itself, their Fig. 2 clearly shows a standing wave pattern of electronic waves reflected

from the system boundary. Our prediction is that extracting the reflection matrix eigenphases from this pattern will yield gapped bands and topological mid-gap states. Note that these static HOTI systems even allow to simulate the stroboscopic time-evolution of Floquet topological phases, as the reflection matrix $r$ of the static system is exactly the stroboscopic evolution operator. Reflecting incoming waves off a HOTI system thus emulates the evolution over a single period. In order to simulate a periodic driving over multiple time periods, our setup has to be altered such that the incoming wave undergoes multiple subsequent scattering processes.

# 7 Conclusion

We have shown that back-scattering from the boundary of a static 2D HOTI is described by a reflection matrix which is topologically equivalent to a 1D nontrivial Floquet system. Depending on the properties of the HOTI and those of the system-waveguide interface, all four phases of the 1D particle-hole symmetric $\mathbb{Z}_2 \times \mathbb{Z}_2$ classification can be obtained. These phases are characterized by quantized topological responses: the phase difference between incoming and outgoing waves is quantized to either 0 or $\pi$ when the modes scatter from the boundaries of the waveguide. Our work introduces a dimensional reduction procedure based on the scattering matrix, which maps the 2D Hamiltonian of a HOTI into the 1D Floquet operator of a nontrivial chain. In Appendix C, we show that this dimensional reduction map remains valid for HOTIs with corner states in any dimension and in any symmetry class, provided the corner states are robust against lattice symmetry breaking. Based on the insight that the reflection matrix can simulate a Floquet operator, we have shown that the topological invariants of the former can be obtained by defining nested scattering matrices.

Our work introduces a novel metamaterial platform: simulating topological phases using reflection matrices. In recent years, topological metamaterials are emerging as an intensely studied research field, both theoretically and experimentally. This is both due to the possibility to accurately control system parameters, as well as in light of potential applications, including signal transmission [83], sensors [84], and even 'light funnels' [85]. By now, topologically nontrivial systems have been experimentally simulated in photonic crystals [86], electronic circuits [40], as well as Josephson junctions [87] and acoustic [41], phononic [38], and mechanical metamaterials [88]. We look forward to the experimental measurements of topology in a reflection matrix.

One of the advantages of using reflection matrices to simulate and measure the properties of Floquet systems is that this method does not suffer from noise-induced decoherence. In noninteracting periodically-driven systems, inevitable noise in the driving field generally leads to exponentially fast decoherence. All topological properties are lost in the long time limit as the system evolves towards a uniform, featureless steady state. The problem of decoherence due to driving noise has been recognized both theoretically and experimentally, starting from early measurements of the quantum kicked rotor in the 90s [89, 90], and an active research field has emerged from studying and attempting to mitigate this effect [26, 27, 91–104]. In our work, we have found a way of eliminating this effect completely: driving noise cannot lead to decoherence if unitary topological phases are realized without any driving.

The results we presented here open several directions for further research also on a theoretical level. First, the dimensional reduction scheme we have introduced might be adapted to reflection matrices which inherit lattice symmetries, such as mirror, rotation, and glide, from the parent HOTI. Building on our work, we anticipate that a link between the classifications of static and driven topological phases can also be established in these systems. Moreover, in three-dimensional HOTIs with corner states, recently realized in acoustic [44–47] and topo-electric [48] metamaterials, we expect the reflection matrix to exhibit corner states of its own,

thus simulating a Floquet HOTI. For three-dimensional systems, it would be interesting to successively apply the nested scattering matrix construction twice. Furthermore, for HOTIs where only two of the four corners show zero modes [105–108], the reflection matrix from one side may show an odd number of topological states, thus simulating a Floquet system which falls outside of the existing, 'tenfold way' [109] classification of topological phases. Finally, it would be interesting to think about how to include many-body effects. While we have treated this system using a purely single-particle, Landauer-Büttiker formalism, the reflection matrix and its topological invariants can be computed also for certain strongly-correlated systems [110–113]. A potential extension of our work would be to see whether interacting Floquet phases, such as the $\mathbb{Z}_3$ parafermion chain of Ref. [114] could be simulated in this way.

## Acknowledgments

We thank Max Geier, Mikael Rechtsman, and Shinsei Ryu for useful discussions, and we thank Ulrike Nitzsche for technical assistance.

**Author contributions**   F.H. and I.C.F. initiated and oversaw the project. S.F. carried out the analysis of the reflection matrix, wrote the code, and performed numerical calculations. S.F. and I.C.F produced the figures. All authors contributed to formulating the results and writing the manuscript.

**Funding information**   This work is supported by the Deutsche Forschungsgemeinschaft (DFG, German Research Foundation) through the Würzburg-Dresden Cluster of Excellence on Complexity and Topology in Quantum Matter – *ct.qmat* (EXC 2147, project-id 39085490) and under Germany's Excellence Strategy – Cluster of Excellence Matter and Light for Quantum Computing (ML4Q) EXC 2004/1 390534769.

## A   Calculating the reflection matrix

Our starting point is the 2D real-space Hamiltonian corresponding to Eq. (1) of the main text. We define two translationally invariant leads along the $x$ direction, where each lead is modeled as an array of isolated waveguides arranged such that each chain connects to only one site of the system. To compute the scattering matrix, we solve the Schrödinger equation [68, 70]

$$(H-E)(\psi_n^{\text{in}} + \sum_m S_{mn}\psi_m^{\text{out}} + \psi^{\text{loc}}) = 0, \tag{6}$$

corresponding to the full, system-plus-leads tight-binding model, described by the Hamiltonian $H$. Here, $E$ denotes the Hamiltonian eigenvalues, $\psi^{\text{loc}}$ stands for wave-functions which are localized in and near the scattering region, and $\psi_n^{\text{in/out}}$ denote incoming and outgoing lead states, that is, plane waves with velocity pointing towards or away from the system, respectively. For detecting states at zero energy, one takes $E = 0$ in the above calculation. The scattering matrix $S$ with elements $S_{mn}$ is obtained directly from the solution of the above equation. All our simulations are done using kwant [70].

In the two terminal geometry, $S$ is a $2 \times 2$ block matrix in which the diagonal blocks are reflection matrices, and the off-diagonal blocks are transmission matrices. We are interested in the regime where the transmission between leads vanishes, yielding unitary reflection matrices $r$ and $r'$. If we construct spinors $\Psi_{L/R}^{\text{in/out}}$ containing all incoming and outgoing modes in both

the left ($L$) and right ($R$) leads, then we obtain

$$\Psi_L^{\text{out}} = r\Psi_L^{\text{in}}, \tag{7}$$

for the reflection matrix of the left lead $r$. The last relation becomes an eigenvalue equation for $\Psi_L^{\text{out}} = e^{i\phi}\Psi_L^{\text{in}}$, implying that the eigenmodes of reflection matrix are standing waves formed as superpositions of $\Psi_L^{\text{in}}$ and $\Psi_L^{\text{out}}$. The spectrum of the reflection matrix, consisting of the eigenphases $\phi$, influences the standing wave pattern formed at the sample boundary.

## B  Symmetries of the reflection matrix and Floquet operator

The classification of topological phases relies on presence or absence of three local symmetries: time-reversal $\mathcal{T} = U_{\mathcal{T}}\mathcal{K}$, particle-hole $\mathcal{P} = U_{\mathcal{P}}\mathcal{K}$ and chiral symmetry $\mathcal{C} = U_{\mathcal{C}}$. Here, $\mathcal{K}$ denotes complex-conjugation, while $U_{\mathcal{T}}, U_{\mathcal{P}}, U_{\mathcal{C}}$ are unitary matrices.

These symmetries constrain the single-particle Hamiltonian $H$ as

$$\begin{aligned}
\mathcal{T}H\mathcal{T}^{-1} = H, \quad \mathcal{P}H\mathcal{P}^{-1} = -H, \quad \mathcal{C}H\mathcal{C}^{-1} = -H, \text{ or} \\
U_{\mathcal{T}}H^*U_{\mathcal{T}}^{\dagger} = H, \quad U_{\mathcal{P}}H^*U_{\mathcal{P}}^{\dagger} = -H, \quad U_{\mathcal{C}}HU_{\mathcal{C}}^{\dagger} = -H.
\end{aligned} \tag{8}$$

Local symmetries of the scattering region and the leads also constrain the scattering matrix $S$ that is related to $H$ via Eq. (6). In order to obtain these constraints using Eq. (6), we need to know the action of these symmetries on incoming modes $\psi_n^{\text{in}}$ and outgoing modes $\psi_n^{\text{out}}$.

Under the action of $\mathcal{T}$ and $\mathcal{C}$, an incoming mode transforms into a linear combination of outgoing modes, and vice versa for the outgoing plane-wave [71]. On the other hand, particle-hole symmetry does not mix incoming and outgoing states. With a unitary matrix $V$ ($W$) denoting the linear combination of outgoing (incoming) plane-waves, the previous statements can be expressed as

$$\begin{aligned}
\mathcal{T}\psi_n^{\text{in}} &= (V_{\mathcal{T}})_{nm}\psi_m^{\text{out}}, \quad \mathcal{C}\psi_n^{\text{in}} = (V_{\mathcal{C}})_{nm}\psi_m^{\text{out}}, \quad \mathcal{P}\psi_n^{\text{out}} = (V_{\mathcal{P}})_{nm}\psi_m^{\text{out}}, \\
\mathcal{T}\psi_n^{\text{out}} &= (W_{\mathcal{T}})_{nm}\psi_m^{\text{in}}, \quad \mathcal{C}\psi_n^{\text{out}} = (W_{\mathcal{C}})_{nm}\psi_m^{\text{in}}, \quad \mathcal{P}\psi_n^{\text{in}} = (W_{\mathcal{P}})_{nm}\psi_m^{\text{in}},
\end{aligned} \tag{9}$$

where implicit summation has been assumed. Time-reversal, particle-hole, and chiral symmetry require that $W_{\mathcal{T}}V_{\mathcal{T}}^* = V_{\mathcal{T}}W_{\mathcal{T}}^* = \mathcal{T}^2 = \pm 1$, $W_{\mathcal{P}}W_{\mathcal{P}}^* = V_{\mathcal{P}}V_{\mathcal{P}}^* = \mathcal{P}^2 = \pm 1$, and $V_{\mathcal{C}}W_{\mathcal{P}} = \mathcal{C}^2 = 1$, respectively. We can choose a basis [33, 71, 108] such that the symmetries act on the lead modes as $U_{\mathcal{T}} = V_{\mathcal{T}}^T = W_{\mathcal{T}}^T$, $U_{\mathcal{P}} = V_{\mathcal{P}}^T = W_{\mathcal{P}}^T$, and $U_{\mathcal{C}} = V_{\mathcal{P}}^{\dagger} = W_{\mathcal{P}}^{\dagger}$, leading to

$$U_{\mathcal{T}}S^*U_{\mathcal{T}}^{\dagger} = S^{\dagger}, \qquad U_{\mathcal{P}}S^*U_{\mathcal{P}}^{\dagger} = S, \qquad U_{\mathcal{C}}S^{\dagger}U_{\mathcal{C}}^{\dagger} = S. \tag{10}$$

In this work, we consider reflections of a 2D HOTI system with gapped bulk and edges. The elements of transmission matrices $t, t'$ are therefore exponentially supressed with system size, yielding unitary $r$ and $r'$. For this reason, Eq. (10) can be reduced to

$$U_T r^*U_T^{\dagger} = r^{\dagger}, \qquad U_P r^*U_P^{\dagger} = r, \qquad U_C r^{\dagger}U_C^{\dagger} = r. \tag{11}$$

The above relations are identical to the symmetry constraints of a unitary Floquet operator:

$$\mathcal{F} = \overline{\exp}[-i/\hbar \int_0^T H(t)dt], \tag{12}$$

where $\overline{\exp}$ denotes the time-ordered exponential and $T$ is the period of the drive. As detailed in Ref. [14], these constraints can be obtained using the definition of the Floquet operator and symmetry relations Eq. (8). They read [14, 63]

$$U_T\mathcal{F}^*U_T^{\dagger} = \mathcal{F}^{\dagger}, \qquad U_P\mathcal{F}^*U_P^{\dagger} = \mathcal{F}, \qquad U_C\mathcal{F}^{\dagger}U_C^{\dagger} = \mathcal{F}. \tag{13}$$

By comparing Eqs. (11) and (13), it is evident that local symmetries constrain $r$ and $\mathcal{F}$ in the same manner. The reflection matrix of a $D$-dimensional HOTI with corner modes in class $S$ is therefore a $(D-1)$-dimensional unitary operator in the same symmetry class $S$.

## C  Dimensional reduction map

In this Appendix we show the conditions under which our dimensional reduction map applies when considering HOTIs of different dimension and symmetry class. We begin by stating these conditions, and then explain them.

**Dimensional reduction map:** Every $D$-dimensional Hermitian TI of order $D \geq 2$, in any symmetry class $S$, maps to a $(D-1)$-dimensional unitary TI of order $(D-1)$, in the same symmetry class $S$, provided that the parent system's topological corner states are robust against lattice-symmetry breaking.

A TI in $D$ dimensions is said to be of order $N$ if it hosts topologically protected, gapless boundary modes of dimension $(D-N)$. The original, strong TIs have an order $N = 1$, so they are called first-order topological phases, since their bulk has dimension $D$ and their gapless surface states have dimension $D-1$. A HOTI with zero-dimensional topological corner states, the focus of our work, is thus a HOTI of order $D$ in $D$ dimensions.

Our dimensional reduction map preserves the symmetry class of the tenfold way classification of Altland and Zirnbauer [109]. This has been shown in the Appendix above by separately treating time-reversal, particle-hole, as well as chiral symmetry. Thus, if a Hermitian system is in symmetry class $S$, the dimensionally-reduced unitary (its reflection matrix) will still be in symmetry class $S$.

This map applies to HOTIs with corner states, in any dimension for which corners can exist (that is $D \geq 2$), as long as they do not rely on lattice symmetry to be protected. This includes all extrinsic HOTI phases, as well as all intrinsic HOTI phases which are converted to extrinsic HOTIs when lattice symmetries are broken. Based on the classification results of [33,34], such $D$-dimensional TIs of order $D$ with $D \geq 2$ are possible in the following symmetry classes: D, DIII, AIII, BDI, and CII.

By now we have clarified the dimensions and symmetry classes which are covered by our dimensional reduction procedure. What is left to show is that the reflection matrix of the $D^{\text{th}}$ order $D$-dimensional Hermitian TI is indeed topological. We achieve this in the following by construction, that is by defining a model indpendent method of obtaining the nontrivial $r$.

Consider a large but finite-sized $D$-dimensional Hermitian TI of order $D$, in a $D$-dimensional hypercube geometry, hosting 0D corner states at its $2^D$ corners. Note that since we consider HOTIs that do not rely on lattice symmetries, such a geometry is always possible, and any such HOTI can be deformed into this geometry after adding suitably many trivial degrees of freedom to its Hamiltonian. Next, place this Hamiltonian in a two-terminal geometry, by connecting two opposite hyper-surfaces to semi-infinite waveguides oriented along one of the $D$ space directions (say $x_1$). The resulting scattering matrix can be computed using Eq. (6). As long as the bulk and all hyper-surfaces are gapped, the transmission between the leads will vanish, and the reflection blocks will be unitary. For concreteness, one can use the same lead Hamiltonian as in the main text, where each of the internal degrees of freedom of each of the boundary sites is connected to a 1D chain with unit hopping and zero onsite term. This has several implications. First, the reflection matrix is $(D-1)$-dimensional, being parameterized by the chain index with real-space coordinates $x_2, x_3, \ldots, x_D$. Second, the open boundary condition will be fixed to the one used in the main text. This means that in the weak coupling limit, waves reflecting far from the corners of the hypercube will pick up a phase $\phi \to 0$. Further, as shown in Refs. [65–68], resonant reflection from the zero-energy corner states of the HOTI

will lead to $\phi = \pi$ for waves localized at the corners of the waveguide.

So far, the construction yields a $(D-1)$-dimensional reflection matrix hosting $\phi = \pi$ modes at each of its $2^{D-1}$ corners. The final step is to show that these corner modes are indeed the consequence of a topologically nontrivial unitary, i.e., they cannot be removed without symmetry-breaking, or closing the gap, or deviating from unitarity. This follows directly from the topological nature of the parent HOTI system. First, the reflection matrix is in the same symmetry class as the HOTI (D, DIII, AIII, BDI, or CII), as discussed in the previous Appendix, which means the corner states are pinned to the middle of the gap [33,34]. The zero-energy corner states of the HOTI, and therefore also the $\pi$ modes of the reflection matrix, cannot be shifted away from this energy (eigenphase) without breaking the local symmetries. The only option left is that the mid-gap modes of the HOTI annihilate pairwise, which must involve a closing of the bulk gap or a closing of some of the hyper-surface gaps of the HOTI. These can have two effects on the reflection matrix: (1) either it stops being unitary, for instance if the HOTI bulk or if hyper-surfaces that connect the two leads become gapless, or (2) it stops having a gap around $\pi$ eigenphase, for instance if the HOTI hyper-surface which becomes gapless is the one contacting the lead. This shows that the reflection matrix is a topological unitary: its $\pi$ modes are protected by unitarity, the eigenphase gap, and the local symmetries.

Finally, we make two remarks about this dimensional reduction map in the context of previous works. First, as shown in Refs. [115, 116], a $D$-dimensional Floquet system can be related to a $(D + 1)$-dimensional static system by defining a so-called 'Floquet Hamiltonian' in a procedure that is commonly called the 'repeated zone scheme.' This is different from our approach, since Refs. [115, 116] explicitly consider periodically-driven systems and the increase of dimension is associated with processes involving the exchange of photons between the system and the driving field. In contrast, our work deals with static systems, in the absence of any driving field. Second, the dimensional reduction procedure defined above is not one-to-one. If it were, this would imply the existence of an inverse, dimensional raising map, which starts from a reflection matrix and produces a higher-dimensional Hermitian HOTI. There is no unique way of doing this, since the reflection matrix encodes only the properties of states close to the Fermi level, whereas the full static HOTI contains also degrees of freedom far from the Fermi level. We note that a lack of invertibility is not a detriment when it comes to establishing connections between topological phases. For instance, the HOTI classification of Ref. [33], as well as the original, 'tenfold way' classification of TIs in Ref. [7] are based on dimensional reduction maps which are not invertible in general. The reason is the same as in our case: the map 'throws away' high-energy degrees of freedom which are unimportant for the topological classification.

## D   Chiral symmetry

It is known that a unitary 1D system with $0/\pi$-modes protected by $\mathcal{P}$ follows a $\mathbb{Z}_2 \times \mathbb{Z}_2$ classification, while a $\mathbb{Z} \times \mathbb{Z}$ classification occurs in 1D systems where $\mathcal{C}$ is the protecting symmetry [14]. To find scattering invariants of a 1D system described by $r$, we calculate the reflection matrix $\tilde{r}(\phi = 0, \pi)$ of one end of this system [63]. For systems with $\mathcal{P}$, the invariant sign $\det[\tilde{r}(\phi = 0, \pi)]$ takes only two values, $-1$ for a system with topologically protected modes and $1$ otherwise. In chiral symmetric phases, the invariant is the number of negative eigenvalues $v_n[\tilde{r}(\phi = 0, \pi)]$ in the basis where $\tilde{r}$ is Hermitian [68].

In the phase with a single $\pi$-mode per end, $\tilde{r}(\phi = \pi)$ is a $2 \times 2$ matrix with $\det[\tilde{r}(\pi)] = -1$. This implies the spectrum of $\tilde{r}(\pi)$ has to be real, and the chiral invariant reads $v_n[\tilde{r}(\pi)] = 1$. Both invariants are thus in agreement. In the phase with a 0-mode at each end, we obtain the same values of invariants for $\phi = 0$. Finally, in the anomalous phase, $\tilde{r}(\phi = 0, \pi)$ is a unitary

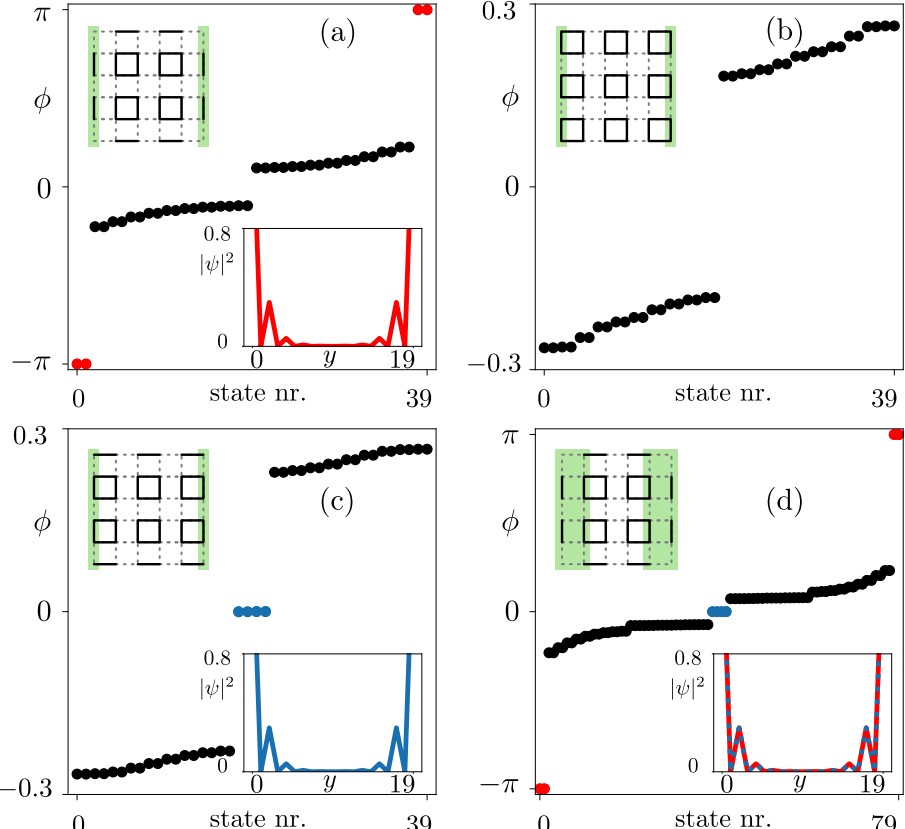

Figure 4: Eigenphases of the reflection matrix, showing $\pi$-modes (0-modes) and their associated wavefunctions in red (blue). We consider a bilayer of HOTI systems, each with $20 \times 20$ sites. In all panels, $\lambda_{x,y} = 1$ and $D_x = D_y = 0.15$. We have chosen $\gamma_{x,y} = 0.4$ in panels (a) and (d), $\gamma_{x,y} = 1.2$ in panel (b), whereas $\gamma_x = 1.2$ and $\gamma_y = 0.4$ in panel (c). Insets sketch the corresponding dimerization pattern of the HOTI as well as the sites connected to the lead (green). The system-lead coupling strength is $t_{sl} = 0.5$ in all cases.

$4 \times 4$ matrix with negative determinant. We transform to the basis in which $\tilde{r}$ is Hermitian via $\tilde{r} \mapsto U\tilde{r}U^\dagger$, $U = \text{diag}(i, 1) \otimes \sigma_0$, and we again obtain $\nu_n[\tilde{r}(\phi = 0, \pi)] = 1$.

For the phases described in the main text, both $\mathcal{P}$ and $\mathcal{C}$ protect only one 0- and/or $\pi$-mode per end. However, by considering two copies of the 2D HOTI system and coupling them in a way that preserves local symmetries, one can obtain a reflection matrix with two modes per end. The momentum-space Hamiltonian for this bilayer system reads

$$H_{\text{bilayer}}(\boldsymbol{k}) = h(\boldsymbol{k})\eta_0 + D_x\tau_y\sigma_z\eta_y + D_y\tau_x\sigma_y\eta_y, \tag{14}$$

where $h(\boldsymbol{k})$ is the Hamiltonian Eq. (1) of the main text. $D_x$ and $D_y$ denote intralayer couplings, while the additional layer degree of freedom is represented by Pauli matrices $\eta$. Here, for simplicity, we assumed intralayer hoppings are momentum independent, i.e., they only connect sites within the unit cell. The chiral symmetry operator reads $\mathcal{C} = \tau_z\sigma_0\eta_0\mathcal{K}$. The calculated reflection matrix spectra for different parameters are plotted in Fig. 4.

In Fig. 4a, the eigenphase spectrum calculated for a bilayer system in the HOTI phase reveals the presence of four $\pi$-modes. They form two pairs, each pair pinned to an end, as shown in the lower inset. Dimerizing a bilayer system trivially in both directions results in a trivial Floquet phase, see Fig. 4b. Furthermore, if the system is nontrivial only in the $y$ direction, the eigenphase spectrum of $r$ contains four 0-modes, as seen in Fig. 4c. Finally,

attaching leads to a full unit cell on the boundary gives an anomalous phase whose spectra is plotted in Fig. 4d. The number of negative eigenvalues of $\tilde{r}$ differs by 2 across the phase transition between trivial and topological phases. For example, the topological phase of a 1D system with $\pi$-modes is described by $\nu_n[\tilde{r}(\phi = \pi)] = 2$ indicating $\nu^\pi = 1$. Therefore, in the bilayer case, the particle-hole invariant cannot distinguish the topological from the trivial phase.

## E   Topological phase transitions of the reflection matrix

The gapless corner modes of a finite 2D system in a HOTI phase, described by the Hamiltonian Eq. (1) of the main text, can depart from zero energy as a result of various gap closings. Some of these phase transitions occur only on the edges, like the $x$-edge gap closing that appears when $\gamma_x = \lambda_x$ and $\gamma_y \neq \lambda_y$ (and vice versa for the $y$-edge). The vanishing of an edge gap is related to the appearance of two counter-propagating modes per edge that are protected by translation symmetry [117–119]. Furthermore, a bulk gap closing occurs for $\gamma_x = \gamma_y = \lambda_x = \lambda_y$ at $(k_x, k_y) = (\pi, \pi)$. These phase transitions affect the reflection matrix $r$ that describes the left edge of the 2D system.

First, we study the topological phase transition of $r$ induced by a $y$-edge gap closing of the 2D system. Here, $r$ remains unitary across the phase transition, provided the HOTI remains insulating in the $x$ direction. Thus, the spectrum of $r$ lies on the unit circle, and henceforth only its eigenphase ($\phi$) spectrum is plotted. For a finite 2D system, the $\phi$-spectrum can have topologically protected states at particle-hole invariant eigenphases, as shown in Fig. 2 of the main text. In this case, the nested scattering matrix topological invariants $\nu^0$ and $\nu^\pi$ can be used to study the phase transition. Here, we consider instead results obtained in a ribbon geometry, for which the HOTI is infinite along the $y$ direction, such that the momentum $k_y$ is a good quantum number. The dispersion of the reflection matrix, $\phi(k_y)$, is shown in Fig. 5 for different values of $\gamma_y/\lambda_y$, while keeping $\gamma_x/\lambda_x < 1$. In this ribbon geometry $r$ is a $2 \times 2$ matrix and its eigenphases form two bands. In Fig. 5a the bands are plotted for a system in a HOTI phase. For the same parameters, the $\phi$-spectrum of an open 1D system has $\pi$-modes (see Fig. 2a of the main text). For $\gamma_y = \lambda_y$ a band gap closing occurs at $\phi = \pi$, thus signaling the hybridization of two $\pi$-modes and their shift into the bulk of the 1D unitary system. Finally, Fig. 5c corresponds to $\gamma_y > \lambda_y$, and shows again two gapped bands. As in Floquet systems, the phase transition of $r$ is accompanied by a gap closing and reopening.

In a similar fashion, one can also study the phase transition between the phase that supports 0-modes and a trivial phase. Then, the dispersion of $r$ would show a band gap closing at $\phi = 0$ for $\gamma_y = \lambda_y$. Finally, $r$ that simulates an anomalous Floquet phase is obtained by attaching lead to a unit cell of sites, and is thus a $4 \times 4$ matrix in the $k_y$-space. The phase transition in this case would involve four bands that cross at both 0- and $\pi$-eigenphases for $\gamma_y = \lambda_y$.

Phase transitions which preserve the unitarity of $r$ can be studied from the perspective of its $\mathbb{Z}_2$ topological invariants $\nu^{0(\pi)}$. By definition, these invariants are discontinuous at the phase transition point, and match the value of $\det[\tilde{r}(\phi = 0, \pi)]$ sufficiently far from it. Thus, in Fig. 6, we plot the dependence of $\det[\tilde{r}(\phi = 0, \pi)]$ on dimerization in the $y$ direction. In Fig. 6a, we start from a system initially dimerized such that $r$ describes a 1D system with $\pi$-modes, hence $\det[\tilde{r}(\pi)] = -1$ and $\det[\tilde{r}(0)] = 1$. The latter quantity does not change across the phase transition, while $\gamma_y/\lambda_y \to 1$ implies $\det[\tilde{r}(\pi)] = 0$. The nested reflection matrix $\tilde{r}$ has a zero eigenvalue due to the conducting $y$-edge [67], while for $\gamma_y > \lambda_y$, $\det[\tilde{r}(\pi)] = 1$. Fig. 6b describes a similar situation, as now we start from a system with 0-modes, and therefore $\det[\tilde{r}(0)]$ changes the value from $-1$ to $1$ across the phase transition. If $r$ describes an

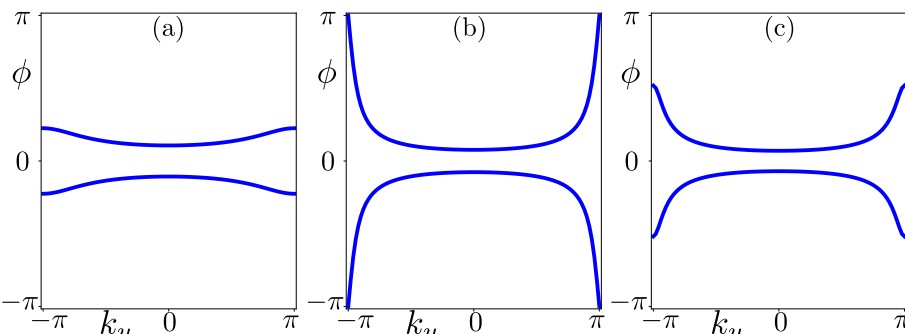

Figure 5: The dispersion $\phi(k_y)$ of the $2 \times 2$ matrix $r$ is calculated across the phase transition related to the $y$-edge gap closing of the 2D system. In all panels $\lambda_x = \lambda_y = 1$, and there are 36 sites in the $x$ direction. In panel (a), the 2D system is in a HOTI phase with $\gamma_x = \gamma_y = 0.4$. The intracell hoppings for panel (b) are $\gamma_x = 0.4$ and $\gamma_y = 1$, and in panel (c) $\gamma_x = 0.4$ and $\gamma_y = 1.2$. The band gap closing at $\phi = \pi$ signals the disappearance of $\pi$-modes in the $\phi$-spectrum corresponding to the reflection matrix of a finite 2D system.

anomalous Floquet phase, increasing $\gamma_y/\lambda_y$ causes both $\det[\tilde{r}(0)]$ and $\det[\tilde{r}(\pi)]$ to change values from $-1$ to $1$, as seen in Fig. 6c.

In the case of an $x$-edge or a bulk gap closing, the finite transmission between the leads renders $r$ subunitary. This implies that not all complex eigenvalues $z$ of $r$ have $|z| = 1$, so not all information is encoded in their phase. Thus, we study these phase transitions by plotting $z$ in the complex plane $(\mathrm{Re}(z), \mathrm{Im}(z))$, like in Fig. 7. For $r$ calculated with open boundary conditions in the $y$ direction, these eigenvalues are represented by black dots. If however, we have a finite system with a periodic boundary condition (PBC) in the $y$ direction, $z$'s are represented by light blue crosses. To visually identify eigenvalues with $|z| = 1$, we always plot a unit circle, colored in grey.

Due to the weak link (hopping strength $t_{\mathrm{sl}}$) between the system and the lead (with hopping $t_{\mathrm{lead}}$), translation symmetry is broken at the system-lead interface. The weak link causes backscattering of the gapless edge and bulk states present in the 2D system at its phase transitions. As such, $\det r = 0$, which would correspond to perfect transmission, does not exactly coincide with gap closings of the 2D system. Rather, $\det r = 0$ for $\gamma_x/\lambda_x = 1 - \delta$, with $\delta > 0$, where $\delta$ is a function of $t_{\mathrm{ls}}$. This explains why in Fig. 7 the zero eigenvalues of the reflection matrix do not occur exactly at $\gamma_x = 1$, but are shifted closer to the value $\gamma_x = 0.925$.

We start with a unitary $r$ whose $\phi$-spectrum has two $\pi$-modes denoted by red color in Fig. 7a. As $\gamma_x$ increases (while keeping $\gamma_y/\lambda_y$ constant), these boundary modes shift along the $x$-axis. For an appropriate $\delta$, they are located exactly at $(\mathrm{Re}(z), \mathrm{Im}(z)) = (0, 0)$ and $\det r = 0$ (see Fig. 7b). By further increasing $\gamma_x/\lambda_x$, these modes move along the horizontal axis to become 0-modes, colored in dark blue in Fig. 7c. Meanwhile, the $r$ eigenvalues calculated in the presence of a PBC remain on the unit circle, as gapless counter-propagating modes on opposite edges are now coupled.

In the proximity of a bulk gap closing, $r$ has a phase transition between a topological and a trivial phase. To study the latter, we start from $r$ with two $\pi$-modes, whose spectrum is plotted in Fig. 7d. As the system approaches the phase transition point, these modes split into a complex-conjugate pair that does not lie on the unit circle, as seen in Fig. 7e. This is because the system conducts but not with a unit conductance due to finite size energy splittings. Finite size effects are eliminated upon the introduction of PBC, as we see two bulk modes at $(\mathrm{Re}(z), \mathrm{Im}(z)) = (0, 0)$ in Fig. 7e. Increasing intracell hoppings with respect to intercell ones moves all modes back to the unit circle. We see in Fig. 7f that $r$ now describes a trivial system.

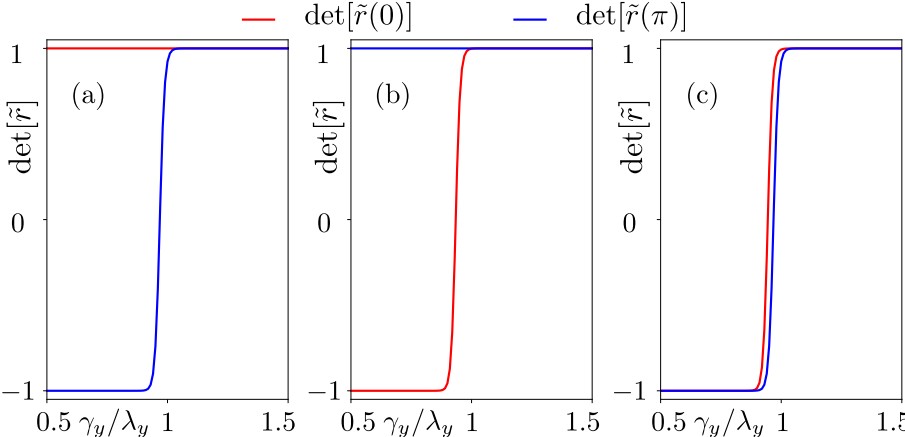

Figure 6: The dependence of $\det[\tilde{r}]$ at eigenphases $\phi = 0, \pi$ on the ratio $\gamma_y/\lambda_y$, with $\lambda_x = \lambda_y = 1$. In panels (a) and (b) we use $\gamma_x = 0.4$ and $\gamma_x = 1.2$, respectively, and $r$ is obtained by attaching leads only to the sites closest to the boundary of the 2D HOTI. In panel (c) however, $r$ is calculated for leads attached to the full unit cell of the 2D system, while the parameters are the same as in panel (a). In all panels, the 2D system consists of $100 \times 100$ sites and it is connected to leads with hopping strength $t_{\text{lead}} = 1$ via a weak link with hopping strength $t_{\text{sl}} = 0.5$.

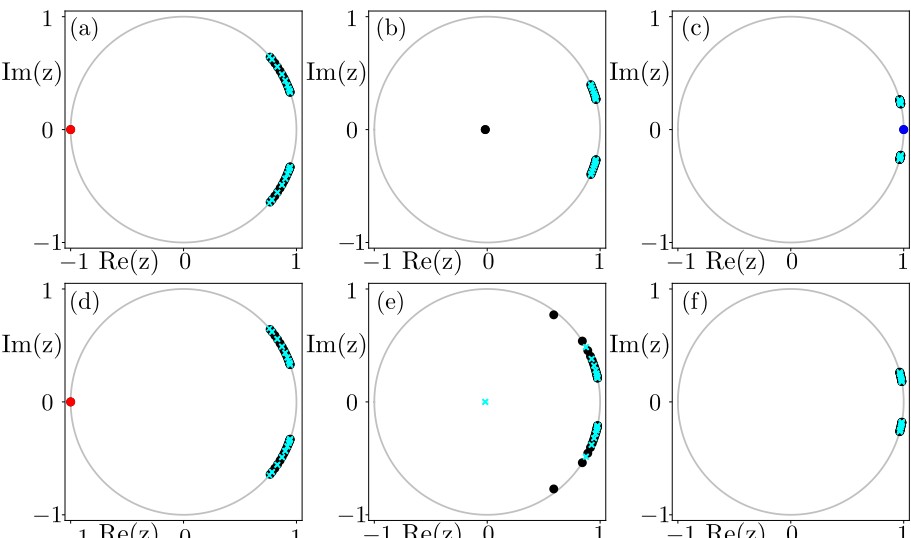

Figure 7: Eigenvalues of $r$ (called $z$) are represented in the complex plane $(\text{Re}(z), \text{Im}(z))$. The matrix $r$ is calculated for a system made of $36 \times 36$ sites. The spectrum of an open system is denoted by black dots, while light blue crosses represent the spectrum in the presence of a periodic boundary condition in the $y$ direction. We always take $\lambda_x = \lambda_y = 1$, and panels (a) and (d) are calculated for $\gamma_x = \gamma_y = 0.4$. Red and dark blue dots correspond to $\pi$-modes and 0-modes, respectively, of the $\phi$-spectrum. Panels (b) and (c) have the same $\gamma_y = 0.4$ and differ in $\gamma_x = 0.925$ and $\gamma_x = 1.2$, respectively. Intracell hoppings in panel (e) are $\gamma_x = 0.925$ and $\gamma_y = 1$, while $\gamma_x = \gamma_y = 1.2$ in panel (f). As in Fig. 6, we take $t_{\text{lead}} = 1$ and $t_{\text{sl}} = 0.5$ for all panels.

# F  Fictitious time-evolution operator

The time-evolution operator for a periodically-driven 1D system described by a Hamiltonian $H(t,k) = H(t+T,k)$ (where $T$ is the driving period) reads

$$U(t,k) = \overline{\exp}[-i/\hbar \int_0^t H(t',k)dt'].\tag{15}$$

Here, $\overline{\exp}$ denotes a time-ordered exponential, and $k$ is the momentum. At $t = T$, $U(T,k) = \mathcal{F}(k)$, where $\mathcal{F}$ is the Floquet operator defined in Eq. (12) of Appendix B.

At $t = 0$, $U(0,k)$ is an identity matrix and all its eigenphases are fixed to zero. One can characterize the Floquet operator topology by studying the winding of the eigenphases of $U(t,k)$ as a function of $t$ and $k$. For a system in class D, the associated topological invariants $Q_0, Q_\pi$ count the number of topological 0-modes and $\pi$-modes at the boundary of the Floquet system. These invariants are computed as the parity of the number of times the eigenphases cross values 0 ($\pi$), in the interval $t \in (0, T]$, at momenta $k = 0$, and $k = \pi$ [72].

In the following, we show how the reflection matrix that simulates a Floquet operator can be continuously deformed to the identity. This deformation simulates a time-evolution process described with a fictitious time-evolution operator. We find a parametrization of the reflection matrix $r(s,k)$ which produces the original reflection matrix for $s = 0$ and the identity matrix for $s = 1$. Note that this corresponds to a 'backwards' time evolution, but yields the same topological invariants. The parameter $0 \le s \le 1$ controls changes in the hopping strengths in Eq. (1).

We start with the scattering region in the limit $\gamma_x = \gamma_y = 0$ and $\lambda_x = \lambda_y = 1$. In real space, there are two $\pi$-modes in the eigenphase spectrum of $r$. In momentum space, the eigenphase spectrum of $r(k_y)$ resembles the one in Fig. 5a and is gapped at $\phi = 0, \pi \,\forall k_y$. To deform it to the identity, we consider a three-step process:

1. For $0 \le s \le \frac{1}{3}$, we change hoppings along the $y$ direction as $\gamma_y = 3s$ and $\lambda_y = 1 - 3s$, with all other parameters kept constant. At the point $\gamma_y = \lambda_y$, there is a $\pi$-gap crossing of the reflection matrix eigenphase bands, like in Fig. 5b. As explained in App. E, this process preserves the unitary nature of the reflection matrix.

2. For $\frac{1}{3} \le s \le \frac{2}{3}$, the dimerization pattern in the $x$ direction alternates, i.e. $\gamma_x = 3s - 1$ and $\lambda_x = 2 - 3s$, with all other parameters kept constant. $r$ remains unitary throughout this process.

3. For $\frac{2}{3} \le s \le 1$, we eliminate the remaining hopping in the $y$ direction by taking $\gamma_y = 3s - 2$. After this step, the 2D system becomes a stack of independent trivial chains oriented in the $x$ direction. The resulting $r = 1$ because every incoming plane-wave will be back-reflected into the lead from the same position and with a vanishing phase difference.

In Fig. 8a and Fig. 8b, we plot the eigenphases of $r(k_y = 0)$ and $r(k_y = \pi)$ during this process. There are no crossings of eigenphases at 0, and thus $Q_0 = 0$ (even parity) is trivial. Indeed, there are no zero-modes at the boundaries of the reflection matrix. For the momentum $k_y = \pi$, the eigenphases close the $\pi$-gap at the point $\gamma_y = \lambda_y = 1/2$ and then evolve towards $\phi = 0$. This implies $Q_\pi = 1$ (odd parity), consistent with the presence of $\pi$-modes at the boundaries of $r$. This $\mathbb{Z}_2$ invariant agrees with the topological invariant $\nu^\pi$ calculated for a finite chain that is obtained using the nested scattering matrix method.

Next, we consider a scattering region in the limit $\gamma_x = \lambda_y = 1$ and $\gamma_y = \lambda_x = 0$. With open boundaries, its reflection matrix supports isolated 0-modes. To deform it to the identity, we consider a following two-step process:

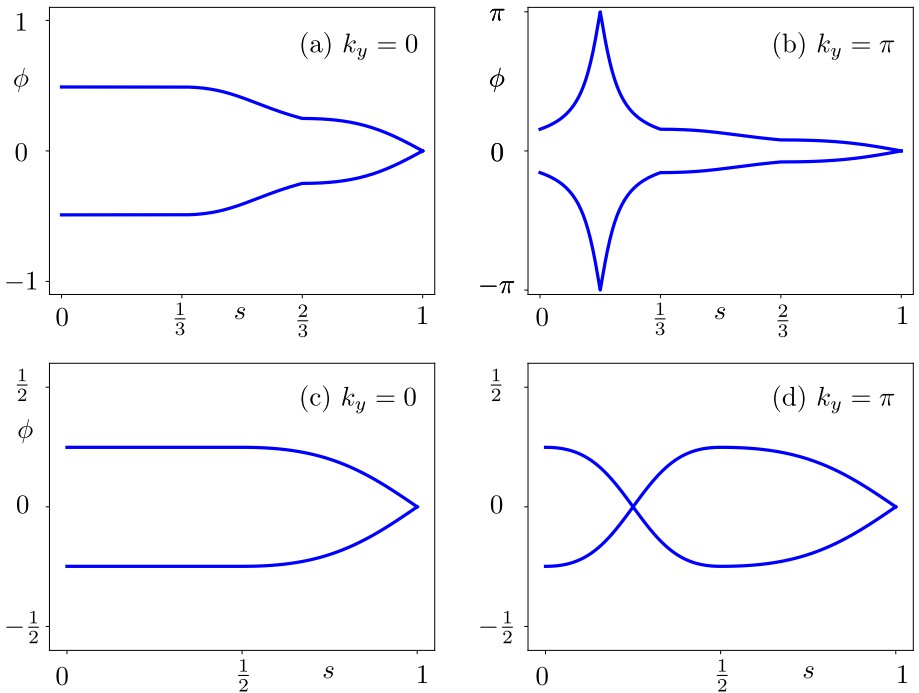

Figure 8: In panels (a) and (b), we consider $r(k_y = 0, \pi)$ that simulates a Floquet system with $\pi$-modes, and show it can be adiabatically related to the identity matrix. In panels (c) and (d), we study how eigenphases of $r(k_y = 0, \pi)$ that simulates a Floquet system with 0-modes change upon smooth deformations to the identity matrix. In panels (a) and (c), we assume $k_y = 0$, and $k_y = \pi$ for panels (b) and (d). In all panels, we consider a system in the ribbon geometry, with 36 sites in the $x$ direction.

1. For $0 \le s \le \frac{1}{2}$, the dimerization pattern in the $x$ direction alternates, i.e. $\gamma_y = 2s$ and $\lambda_y = 1 - 2s$, with all other parameters kept constant. As before, $r$ remains unitary throughout this process.

2. For $\frac{1}{2} \le s \le 1$, we eliminate the remaining hopping in the $y$ direction by taking $\gamma_y = 2 - 2s$. Following this step, the 2D system is a stack of independent trivial chains oriented in the $x$ direction, and its $r = 1$ as explained previously.

Finally, we show in Fig. 8c (Fig. 8d) how the eigenphases of $r(k_y = 0)$ ($r(k_y = \pi)$) change during this two-step process. We observe no crossing at eigenphase $\pi$, so $Q_\pi = 0$ (even) is trivial, consistent with the absence of $\pi$ modes at the boundaries of the reflection matrix. There is only one crossing at zero eigenphase, such that the invariant $Q_0 = 1$ (odd) agrees with the $\mathbb{Z}_2$ invariant $\nu^0$ calculated for a finite chain.

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
