# Peer review of "Simulating Floquet topological phases in static systems"

_SciPost Physics, doi:SciPost Phys. Core 4, 007 (2021)_

## Round 1 · Referee Report · Anonymous · 2021-2-24

Report

Following the comments given in the previous submission, the authors included a more detailed general description of their dimensional reduction procedure and compared it with another competing approach.

This is clearly an improvement from the previous manuscript hence I now vote to accept.

Regarding the discussion around the importance of simulating non-interacting physics, my opinion stayed as is.

---

## Round 1 · Author Response

Dear Editor,
Thank you for handling our manuscript and for sending us the referee reports.
For ease of reading, we have prepared a pdf reply, which can be accessed at the following link:

https://www.dropbox.com/s/3xpait2dc0u7fa9/reply_09_02_2021.pdf?dl=0

Sincerely,
Selma Franca, Fabian Hassler, and Ion Cosma Fulga

---

## Round 1 · List of Changes

See our reply pdf for a detailed list of changes.

---

## Round 2 · Referee Report · Anonymous · 2020-11-10

Strengths
1. The paper offers a new perspective on the topological nature of HOTI boundary phenomena, that of scattering matrices.
2. It may, with the reservation below, lead to interesting connections between the classification of HOTIs and Floquet systems.
3. It provides a new, although somewhat indirect, way for experimenting with Floquet topological phases.
Weaknesses
1. The mapping between HOTI experiments and Floquet experiments is somewhat artificial and indirect. If one is truly interested in witnessing Floquet related effects, one can argue that simulating Floquest systems on a computer has similar value to the "analog" simulation technique presented here.
2. It does not generalize in any trivial manner to interacting systems where richer physics can emerge.
3. The paper envisions a physically-inspired mapping/dimensional-reduction between static topological phenomena and Floquet, via scattering matrices. However, it is not clear to me that such a mapping can exist. For instance, I believe that scattering of the HOTI edge considered in that work would coincide with scattering from (the bulk and boundary of) an SSH chain. The authors should clarify whether this envisioned mapping between static and Floquet phenomena can indeed be 1 to 1.
Report
The current work studies the reflection of a boundary of a 2d HOTI and shows that the topological nature of the edge is reflected in the spectrum of the back-scattering matrix. This is then suggested as a means to simulating topological Floquet systems, and also as a new dimensional reductions scheme between topological phenomena in different dimensions.
I'm somewhat on the fence with this work. The dimensional reduction technique suggested is novel as far as I know. However the authors, in my mind, do not describe the detail and potential of such a mapping in sufficient detail. It is clear, to some extent, that some topological features of the edge state should leave signatures on the reflection matrix but are these signatures or a real mapping between two seemingly different topological phenomena?
An aspect of the work which is promoted more strongly concerns the simulation of topological Floquet systems. However, this way of experimenting with Floquet system seems somewhat indirect to me and also limited to non-interacting systems where simulation techniques are more abundant, most notably on a computer.
Given these two points, I believe this work falls slightly below the acceptance threshold. However, I'd reconsider this if the authors flesh out their dimensional reduction technique in more detail (see requested changes below).
Requested changes
1. As a test of their dimensional reduction approach, can the authors show a matching between classification tables of Floquet topological phases and HOTIs? One which, obviously, translates the symmetries in a clearly prescribed manner.
Anonymous on 2021-01-07 [id 1127]
The manuscript entitled 'Simulating Floquet topological phases in static systems' attempts to offer a new platform for realizing Floquet topological phases on edges of static higher-order topological phases. While the idea appears to be new, my concern lies with how specific the model is to the kind of models considered here (free-fermions) and its limited significance since such models, driven or otherwise, can be simulated quite efficiently. Moreover, there seems to be a fair amount of conceptual issues that have not been touched upon in this work, either in the way of connecting to previous work or distinguishing the current one from them.
Let me elaborate pointwise: 1. The authors use the fact that the reflection matrix on a d-1 edge of a higher-order TI (with gapless topological states at the d-2 corners) is analogous to a d-1 Floquet unitary with topological modes at zero or pi quasienergy (or both). However, it is not at all clear from the manuscript how would that translate to some non-trivial response, such as the frequency dependence of spectral functions, characteristic of non-trivial Floquet phases? 2. The non-trivial topology of the reflection matrix comes about due to the pi phase picked up at the corner states. How does this generalise to richer and arguably, more non-trivial Floquet phases, such as those in Z_3 parafermion chains [Phys. Rev. B 94, 045127 (2016)]? 3. The authors use a specific invariant based on scattering matrices to characterise the topology of the reflection matrix and say that they cannot use the usual winding invariants used for Floquet unitaries as they dont have access to the instantaneous eigenmodes. This begs the question that whether the results presented in the manuscript can be interpreted as genuine Floquet topological modes or just a constructed unitary which mimicks them. After all, the non-trivial topology in the latter is due to how the quasienergies wrap around the quasienergy-momentum Brillouin zone. 4. Regarding the issue of dimensional reduction, it is well known that a d-dimensional Floquet system can be mapped onto a (d+1)-dimensional static system [Phys. Rev. 138, B979 (1965); Phys. Rev. A 7, 2203 (1973)]. Is the idea of dimensional reduction described in this work related to the aforementioned old works? If yes, then the authors should comment on the connection; if not then how is their idea different? 5. Finally, the authors comment that one benefit of their work is that it does not suffer from decoherence or heating due to driving. I am not sure of the significance of this comment since free-fermionic systems anyway do not heat up and studying Floquet topological phases therein does not suffer from that problem. The authors claim that in reality, driving is noisy and it could lead to heating. This comment is somewhat unfair as at the end of the day, these are all model studies, and in reality fermions aren't non-interacting either.
Overall, I think the manuscript is well-written but has quite a few gaps pertinent to earlier concepts on Floquet systems as well as analysis of the current model. However, the major reason why I think the manuscript in its present form is not suitable for SciPost is due to the limited applicability and significance of the results.

---

## Round 3 · Referee Report · Anonymous · 2021-2-24

Report
Following the comments given in the previous submission, the authors included a more detailed general description of their dimensional reduction procedure and compared it with another competing approach.
This is clearly an improvement from the previous manuscript hence I now vote to accept.
Regarding the discussion around the importance of simulating non-interacting physics, my opinion stayed as is.
Anonymous on 2021-03-23 [id 1328]
Following the Referee reports and the comments on the previous version of the manucript, the authors have made modfications to the manuscript, which I think has improved it quite a lot. In particular, the appendices on dimensional reduction and the fictitious time-evolution operator go a long way in improving the paper.
I find the statement of the authors' about a response (such as in frequency-dependence of spectral functions) unconvincing. Even if the the 'invariant' is a pi phase difference in the unitary reflection matrix, can it not be engineered into a signature of the topology such as in 4pi Josephson junctions?
The authors make several claims of meta-metarial realization of their protocol; however I do not think it is substantiated well. I understand that this is a theoretical paper and the scope for a full experimental proposal is limited.
The concern about the significance of the results stays unchanged.
Despite these concerns, with some reservations, I think the paper can be considered for SciPost Physics Core.
Jian-Hua Jiang on 2021-03-11 [id 1296]
This is definitely an elegant theory that connects two realms of topological physics: the Hermitian systems and the unitary systems. The mapping looks interesting and encouraging. I am wondering how such a connection depends microscopic details: for instance, the size of the HOTI, the couplings between the waveguides and the HOTI, the band gap of the HOTI. In addition, there are various 2D HOTIs, how do they map to 1D Floquet topological insulators in various phases? All these aspects are important before the phenomena become reality in experiments.
Author: Ion Cosma Fulga on 2021-04-15 [id 1364]
(in reply to Jian-Hua Jiang on 2021-03-11 [id 1296])Thank you very much for your comment! We have replied to it in the attached pdf.
Attachment:
reply.pdf
Ion Cosma Fulga on 2021-02-09 [id 1219]
Following the Editor's suggestion, we upload here the pdf of our resubmission letter.
Attachment:
reply_09_02_2021.pdf

---

## Round 3 · Author Response

Thank you for handling our manuscript and for sending us the referee reports.
For ease of reading, we have prepared a pdf reply, which can be accessed at the following link:
https://www.dropbox.com/s/3xpait2dc0u7fa9/reply_09_02_2021.pdf?dl=0
Sincerely,
Selma Franca, Fabian Hassler, and Ion Cosma Fulga

---

## Round 3 · List of Changes

See our reply pdf for a detailed list of changes.

---

## Editorial Decision

published